# Computational Analysis of SAM Analogs as Methyltransferase Inhibitors of nsp16/nsp10 Complex from SARS-CoV-2

**DOI:** 10.3390/ijms232213972

**Published:** 2022-11-12

**Authors:** Alessandra M. Balieiro, Eduarda L. S. Anunciação, Clauber H. S. Costa, Wesam S. Qayed, José Rogério A. Silva

**Affiliations:** 1Laboratório de Planejamento e Desenvolvimento de Fármacos, Instituto de Ciências Exatas e Naturais, Universidade Federal do Pará, Belém 66075-110, Brazil; 2Institute of Chemistry and Center for Computing in Engineering & Sciences, University of Campinas, Campinas, São Paulo 13084-862, Brazil; 3Medicinal Chemistry Department, Faculty of Pharmacy, Assiut University, Assiut 71526, Egypt

**Keywords:** SARS-CoV-2, nsp16/nsp10, SAM analog, inhibition mechanism, MD simulations, binding free energy

## Abstract

Methyltransferases (MTases) enzymes, responsible for RNA capping into severe acute respiratory syndrome coronavirus 2 (SARS-CoV-2), are emerging important targets for the design of new anti-SARS-CoV-2 agents. Here, analogs of S-adenosylmethionine (SAM), obtained from the bioisosteric substitution of the sulfonium and amino acid groups, were evaluated by rigorous computational modeling techniques such as molecular dynamics (MD) simulations followed by relative binding free analysis against nsp16/nsp10 complex from SARS-CoV-2. The most potent inhibitor (**2a**) shows the lowest binding free energy (–58.75 Kcal/mol) and more potency than Sinefungin (SFG) (–39.8 Kcal/mol), a pan-MTase inhibitor, which agrees with experimental observations. Besides, our results suggest that the total binding free energy of each evaluated SAM analog is driven by van der Waals interactions which can explain their poor cell permeability, as observed in experimental essays. Overall, we provide a structural and energetic analysis for the inhibition of the nsp16/nsp10 complex involving the evaluated SAM analogs as potential inhibitors.

## 1. Introduction

In October 2022, the World Health Organization (WHO) reported over 600 million cases and over six million deaths since the beginning of the COVID-19 pandemic [1]. This disease is caused by the severe acute respiratory syndrome coronavirus 2 (SARS-CoV-2), an enveloped β-coronavirus formed by a large and complex positive-sense single-stranded RNA genome [2]. Particularly, these viruses have one of the largest genomes of all RNA viruses. For SARS-CoV-2, its genome has ~29,800 bases, which are responsible encodes a total of 4 structural and 16 nonstructural proteins (named nsp1–nsp16), which are crucial for the virus survival [3,4]. Among these potential targets, some nonstructural proteins (nsps) which play key roles in RNA capping in coronavirus [5,6,7] have been used for drug design of anti-SARS-CoV-2 agents [8,9,10,11], once that RNA cap modification contributes to host cell defense as viral RNA lacking 2’-O methylation which is sensed and inhibited by the interferon-stimulated IFIT-1 protein [12].

Previous studies involving human and animal coronaviruses have demonstrated that the nsp16/nsp10 complex is responsible for the Cap-0 binding of mRNA due to its (Nucleoside-2′O)-methyltransferase activity [13]. Particularly, the activity of nsp16 is improved by the presence of nsp10, which plays a cofactor rule [8,14]. In the Cap-0 reaction mechanism, the nsp16 methylates the mRNA by transferring its methyl group from S-adenosylmethionine (SAM) donor to the unmethylated ribose 2’-O, obtaining RNA-2′-O-methylated and S-adenosyl homocysteine (SAH) [15,16].

Recently, Bobileva and co-workers [11] synthesized analogs of SAM as inhibitors of viral mRNA cap methyltransferases (MTases). They evaluated these new inhibitors into nsp14 and nsp16/nsp10 from SARS-CoV-2 and human glycine N-methyltransferase (GNMT), where five compounds show nanomolar to submicromolar IC_50_ values. However, these compounds did not show selectivity concerning human GNMT and have poor cell permeability. Therefore, understanding the inhibition mechanism of the nsp16/nsp10 complex is important for elucidating molecular details which can explain the source of no selectivity and poor cell permeability. Some computational studies have indicated that the origin of the catalytic effect of methyltransferases is mainly due to electrostatic preorganization [17,18,19,20,21]. Recently, our group has described the catalytic mechanism of the 2′-O methylation of the viral mRNA cap by applying Quantum Mechanics/Molecular Mechanics (QM/MM) with MD simulations [9]. Besides, recent computational studies have been published involving inhibitors of 2’-O-Methyltransferase from SARS-CoV-2 [22,23,24]. Overall, we aim to understand the determinants of nsp16/nsp10 inhibition by SAM analogs and provide insights into their poor cell permeability by performing powerful computational analysis.

## 2. Results and Discussion

### 2.1. Structural Analysis of MD Simulations and PCA Analysis

The structural stabilization for all nsp16/nsp10 systems is evaluated by using RMSD and RMSF plots computed by the CPPTRAJ program [25] (Figure 1A,B). In Figure 1A, the simulated nsp16/nsp10 systems show suitable convergence considering their respective average structures. These observations corroborate the computed RMSDs values: 1.4 ± 0.3, 1.7 ± 0.2, 1.7 ± 0.1, 2.6 ± 0.1, 1.6 ± 0.1 and 2.2 ± 0.1 Å for **1a**, **2a**, **2b**, **4c**, SAM and SFG systems, respectively. These results suggest that the SAM and its analogs systems were well equilibrated during MD simulation. Interestingly, **1a**, **2a** and **2b** systems show RMSD values close to the SAM system, while SFG and **4c** systems show the highest and most similar RMSD values (2.2 ± 0.1 and 2.6 ± 0.1 Å, respectively). From the RMSF analysis (Figure 1B), the values were: 0.7 ± 0.6, 0.8 ± 0.8, 0.7 ± 0.7, 0.7 ± 0.6, 0.7 ± 0.6 and 0.7 ± 0.8 Å for **1a**, **2a**, **2b**, **4c**, SAM and SFG systems, respectively. Although their respective standard deviations are on the same scale, mainly due to changes in the terminal region, the protein section which comprises amino acid residues from Asp125 to Phe150 shows a significant difference, which is more evident for Thr140 (2.4 and 2.5 Å for SAM and **2a**, respectively). These structural analyses suggest that even small differences can be determinants for the binding of all evaluated compounds.

To better understand the collective and individual movements of the nsp16 structure, we applied PCA analysis for all simulated systems. It is also important to highlight that the nsp10 protein (cofactor) was not included in the PCA analysis, the discussion focus is on the nsp16 protein. All 200 ns of MD simulations were considered to obtain the protein movement, all analyzed systems were plotted using three combinations of the principal components (PCs): PC1 vs. PC2 (Figure 2), PC1 vs. PC3, and PC2 vs. PC3. PC plots for **1a**, **2b,** and **4c** systems are provided in the Appendix A.

As can be observed in Figure 2, where the progress of the trajectory for all nsp16 protein systems is shown, each point of the plot means movement direction for the nsp16-SAM, nsp16-SFG, and nsp16-**2a** during the 200 ns of MD simulation. The clustering of the conformers in the PC1 vs. PC2 plot in SAM and SFG systems have a similar profile, which means similar conformational behavior during MD simulations, which could be associated with a small motion of the nsp16 structure. However, the **2a** system shows a different profile for conformational changes during MD simulations, which can suggest induction caused by the **2a** inhibitor into the catalytic site of nsp16.

As can be seen in Figure 3 and Table 1, for SAM and SFG systems residues Asp99, Asp114 and Asp130 maintain relevant hydrogen bonds with donor atoms from SAM and SFG structures. Particularly, Asp99 shows an occupancy percentage greater than 100%, it occurs due to the carboxylic group of Asp99 interacting with both hydroxyl groups of the furan ring present in SAM and SFG structure, which allow a suitable orientation for the catalytic reaction of nsp16/nsp10 complex [9]. Interestingly, the Asp130 residue does not show any hydrogen interaction with synthesized SAM analogs due to the absence of the aminobutanoate group in their structures. These results can provide a clue about the structural effects caused by modifications to SAM analogs, which will corroborate our energetic analysis.

### 2.2. Binding Free Energy and Residual Decomposition Analysis

As described in the Material and Methods section, relative binding free energies for all nsp16/nsp10 systems were performed using the MM/GBSA method [26,27,28] as implemented in *MMPBSA.py* [29] and are presented in Table 2. It should be highlighted that a single MD trajectory of the bound complexes was considered to compute the relative binding free energy ($\Delta G_{bind}$), where the $\Delta E_{int}$ term (in Equation (3)) is canceled due the energy differences are from the same MD ensemble [27,28].

The thermodynamic terms related to the binding free energy are listed in Table 2. As can be observed, van der Waals ($\Delta E_{vdW}$) and non-polar ($\Delta G_{SA}$) terms are the most consistent components for binding free energy, ranging from −51.2 to −41.8 Kcal/mol for $\Delta E_{vdW}$ and from −6.8 to −5.9 Kcal/mol for $\Delta G_{SA}$. The calculated binding free energy $(\Delta G_{bind}$) values for SAM and SFG systems are −70.0 and −39.8 Kcal/mol, respectively, suggesting a most favorable bind for SAM, the natural substrate of the nsp16/nsp10 complex. As can be observed for the selected SAM analogs (**1a**, **2a** and **2b)**, they have a lower $\Delta G_{bind}$ (−46.8, −58.7 and −49.6 Kcal/mol, respectively) than the pan-MTase inhibitor (SFG, −39.8 Kcal/mol) which agrees with experimental evidence found by Bobileva et al. [11]. According to Bobileva et al. [11] compound **4c** showed minimal activity (IC_50_ = 223 µM), the $\Delta G_{bind}$ found was −30.1 Kcal/mol, the highest value among all simulated compounds. On the other hand, compound **2a** has the lowest $\Delta G_{bind}$ (−58.7 Kcal/mol) which agrees with the results found by Bobileva et al. [11]. Therefore, our applied computational strategy described the same binding tendency observed by Bobileva et al. [11] for the most potent compounds (**1a**, **2a** and **2b**) as well as for the inactive compound (**4c**).

Interestingly, the electrostatic ($\Delta E_{ele}$) and polar ($\Delta G_{GB}$) term showed significant differences between SAM and SFG related to other compounds. For $\Delta E_{ele}$ values of SAM and SFG, we can see an increase from −370.4 to −83.5 Kcal/mol, respectively. However, the increase of $\Delta E_{ele}$ values for **1a**, **2a** and **4c** (72.0, 34.8 and 98.8 Kcal/mol, respectively) are more evident. Among selected compounds, only 2b shows a negative value for $\Delta E_{ele}$ (−54.6 Kcal/mol). About the polar term, SAM, SFG and **2b** show positive values (358.4, 94.8 and 63.2 Kcal/mol, respectively), while **1a**, **2a** and **4c** compounds show negative values (−71.0, −38.8 and −74.9 Kcal/mol, respectively). Our results suggest that $\Delta E_{ele}$ and $\Delta G_{GB}$ terms could be related to the cell permeability presented by SAM analogs in A549 cell line essays performed by Bobileva et al. [11].

A plot of a residual decomposition analysis of $\Delta G_{bind}$ (Figure 4) was included to improve the energetic description of the features that contributed to the recognition and binding in all the nsp16 systems. In Table 3 any residue with values below −1.20 Kcal/mol was included as an important residue in the binding process.

As can be observed in Table 3, some important residues for the SAM binding as Tyr47, Gly73, Asp99 and Asp130 (−2.2, −1.8, −12.7 and −4.4 Kcal/mol, respectively) decrease significantly for SFG (0.0, −1.2, −6.1 and 0.3 Kcal/mol, respectively) and in others SAM analogs. Particularly, the carboxylic group of Asp99 maintains strong H bonds with both hydroxyl groups of the furan ring of SAM, which explain the high occupancy values shown in Table 3. By another hand, some residues, such as Asn43, Ser74, Lys76 and Met131 appear to be important to the binding of SFG and synthesized SAM analogs. Interestingly, for the **2a** compound, Lys76 has the lowest interaction value (−10.4 Kcal/mol) among all computed systems, compensating for the increased interaction with Asp130, which indicates its great importance for the binding of the most active synthesized SAM analog, in agreement with Bobileva et al. results [11]. Our computational results suggest that modifications performed by Bobileva et al. [11] to the SAM analogs, although, have allowed new interactions with the catalytic site of nsp16, important interactions (e.g., Asp99 and Asp130) were drastically decreased according to MD and free energy analysis.

## 3. Computational Methods

### 3.1. System Setup and MD Simulations

Initially, we have chosen the nsp16/nsp10-SAM complex interacting with RNA from PDB code 6WKS [8] and well equilibrated from our previous nsp16/nsp10 study [9]. The 3D structures SAM and its analogs (SFG, **1a**, **2a**, **2b** and **4c**) (Figure 5) were in silico build and structurally minimized at the quantum mechanics (QM) level by applying the Hartree–Fock (HF) method with a 6-31G ** basis set and the RESP method [30] was used for the partial charges calculations carried out in the Gaussian09 package [31]. To estimate the protonation states of the titratable amino acid residues, a pKa calculation was performed using the PROPKA method [32] at pH 7. The ff14SB [33] was applied for the protein (nsp10 and nsp16) while GAFF [34] was used for the RNA part and SAM and its analogs. It is important to highlight that nsp10 is a zinc-binding protein, then Zinc AMBER force field (ZAFF) [35] parameters were used for the description of its metal center containing Zn^2+^ ions. Then, *tleap* module of the Amber20 program [36] was used to add protons of the protein (nsp10 and nsp16). Each system (nsp16-nsp10-X-m7GpppA-RNA; “X” means ligand) was immersed in a truncated octahedral cell of water molecules described by TIP3P [37] model, extending 8 Å away from the solute part. All technical procedures were detailed previously [9]. In all simulation stages, the particle mesh Ewald (PME) method was used to calculate the long-range electrostatic forces employing a nonbonded cutoff of 10 Å, and H-bonds were constrained by using the SHAKE method [38]. The PMEMD module of the Amber20 program [36] was used for all MM simulations.

### 3.2. Structural and Thermodynamic Analysis

The root-mean-square deviation (RMSD) and root-mean-square fluctuation (RMSF) of the backbone atoms (Cα, N, O, C) plots were computed to avail structural stabilization of all simulated systems, where the trajectories ensembles were fitted to the average structures from production stages by using the CPPTRAJ module [25]. Besides, the principal component analysis (PCA) approach [39,40] was applied to explore the local and collective movements of nsp16/nsp10 systems that occurred during the MD simulations

Thereafter, the CPPTRAJ module [25] was used to select 20 ns (a total of 2000 representative snapshots) from the production stage of the MD simulations for the binding free energy ($\Delta G_{bind}$) calculations using the MM/GBSA approach [26,27,28], as implemented into the *MMPBSA.py* module [29] of AmberTools20. The main equations of the $\Delta G_{bind}$ by MM/GBSA are computed as follows:(1)$$\Delta G_{bind}=\Delta H-T\Delta S=\Delta E_{MM}+\Delta G_{SOLV}-T\Delta s$$
(2)$$\Delta E_{MM}=\Delta E_{int}+\Delta E_{ele}+\Delta E_{vdW}$$
(3)$$\Delta G_{SOLV}=\Delta G_{GB}+\Delta G_{SA}$$
where $\Delta G_{bind}$ is computed from the gas-phase MM energy ($\Delta E_{MM}$), solvation energy ($\Delta G_{SOLV}$) and the entropic term (−TΔS) (Equation (1)). The $\Delta E_{MM}$ is the sum of the changes in the internal (bond, angles, and dihedral energies) ($\Delta E_{int}$), electrostatic ($\Delta E_{ele}$) and van der Waals ($\Delta E_{vdw}$) interactions (Equation (2)). As a single-trajectory scheme is applied for the $\Delta G_{bind}$ calculations, the $\Delta E_{int}$ is equal to zero. The $\Delta G_{SOLV}$ includes the polar ($\Delta G_{GB}$) and non-polar ($\Delta G_{SA}$) energies for $\Delta G_{bind}$ (Equation (3)). Due to the computational cost, the entropic term (−TΔS) was not included in the $\Delta G_{bind}$ calculations [27,28]. Furthermore, a per-residual decomposition analysis was computed to provide insights into the relative contribution of the amino acid residues [26]. This method has been successfully used in SARS-CoV-2 drug design studies [9,41,42,43,44,45,46,47].

## 4. Conclusions

In this study, we have used MD simulations followed by structural analysis and binding free energy calculations to evaluate the key features of the inhibition of nsp16/nsp10 complex from SARS-CoV-2 by the SFG and new SAM analogs. Our results are in good agreement with the experimental evidence proposed by Bobileva et al. [11]. The most active compound (**2a**) shows the lowest binding free energy value (−58.7 Kcal/mol) among all SAM analogs, including the reference inhibitor, SFG. Interestingly, the significant interaction of **2a** with Lys76 (−10.4 Kcal/mol) can suggest it is a key residue for the binding of this SAM analog and its good activity. Regarding the poor cell permeability results found for these new SAM analogs, our results suggest that it could be related to the positive values of electrostatic interactions computed by the MM/GBSA calculations. Finally, the insights provided by applied computational techniques here may be used as leads for further drug development based on SAM analogs as inhibitors of MTases from SARS-CoV-2.

## Figures and Tables

**Figure 1 ijms-23-13972-f001:**
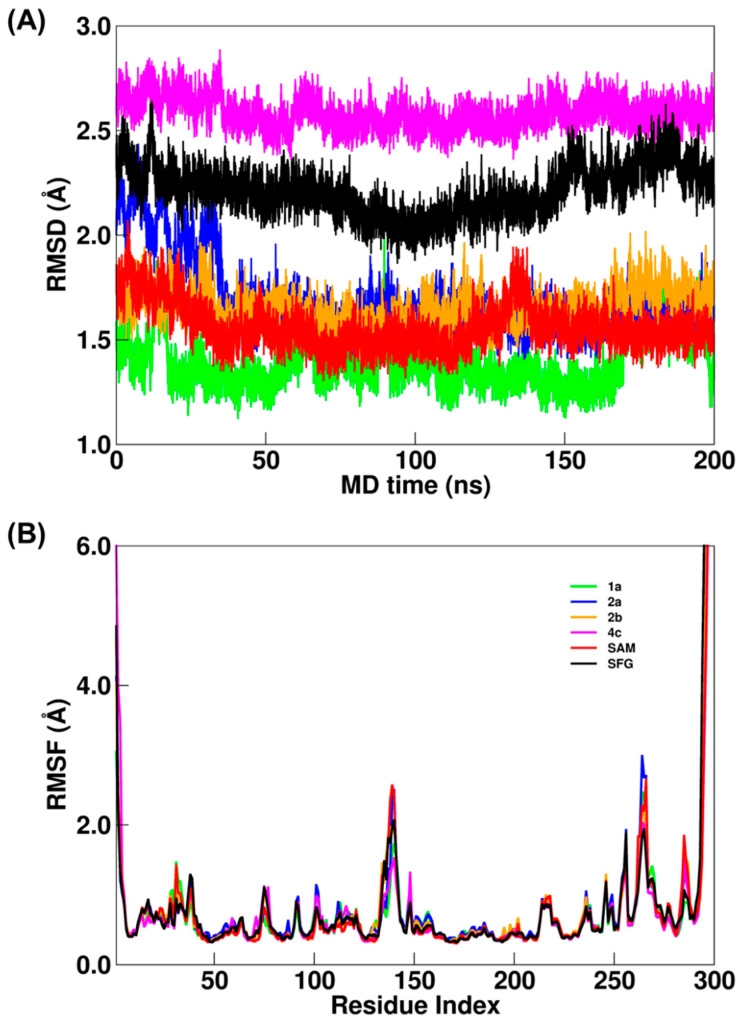
(**A**) RMSD and (**B**) RMSF plots for nsp16-**1a** (green), nsp16-**2a** (blue), nsp16-**2b** (orange), nsp16-**4c** (magenta), nsp16-SAM (red) and nsp16-SFG (black) systems. All values are reported in Å. RMSF values are included as Appendix A.

**Figure 2 ijms-23-13972-f002:**
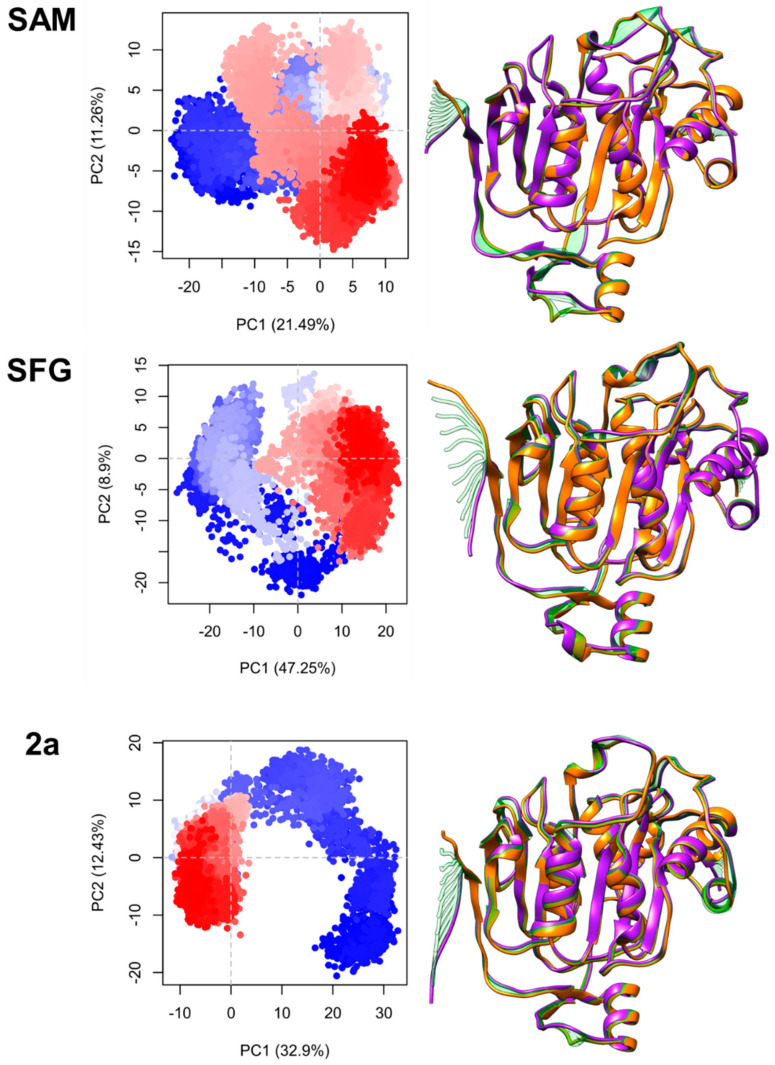
Conformations acquired by the nsp16 protein during 200 ns of MD simulations for SAM, SFG, and **2a** systems. PCA plots (left) from the initial (blue) to 200 ns (red) structures. Initial, intermediate, and final 3D conformations are highlighted in orange, green, and purple, respectively.

**Figure 3 ijms-23-13972-f003:**
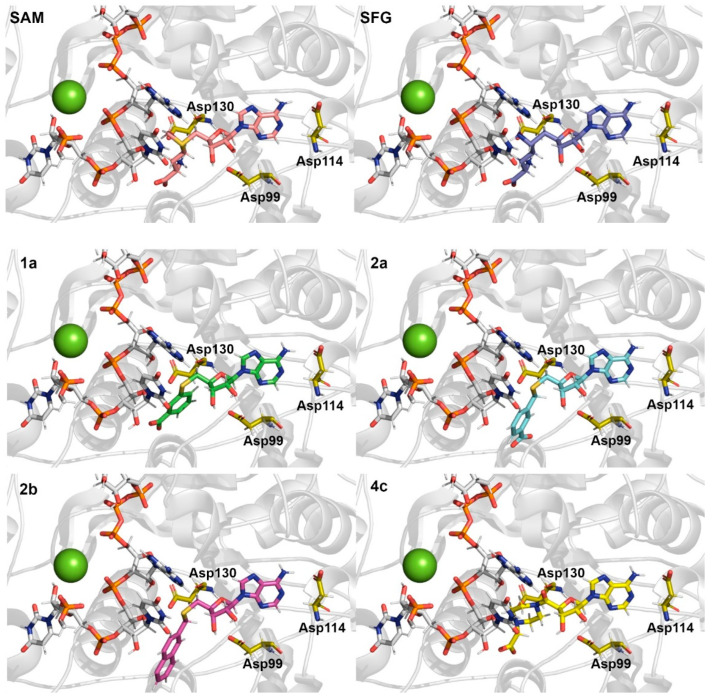
Relevant Asp residues interact by H bonds with SAM (pink color), SFG (dark blue color), **1a** (green color), **2a** (cyan color), **2b** (purple color) and **4c** (yellow color). The Mg^2+^ ion is shown as a green sphere and the RNA part is shown as a stick model (C atoms in white color). The PBD file for each one is provided as Appendix A.

**Figure 4 ijms-23-13972-f004:**
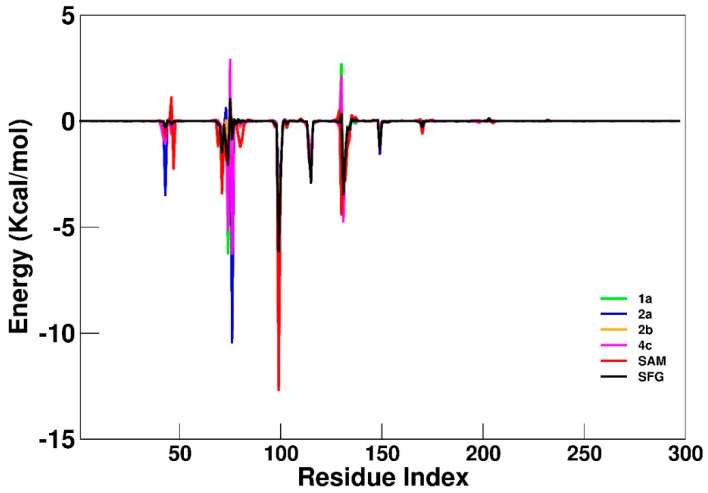
Residual decomposition plot for nsp16-**1a** (green), nsp16-**2a** (blue), nsp16-**2b** (orange), nsp16-**4c** (magenta), nsp16-SAM (red) and nsp16-SFG (black) systems. All values are reported in Kcal/mol.

**Figure 5 ijms-23-13972-f005:**
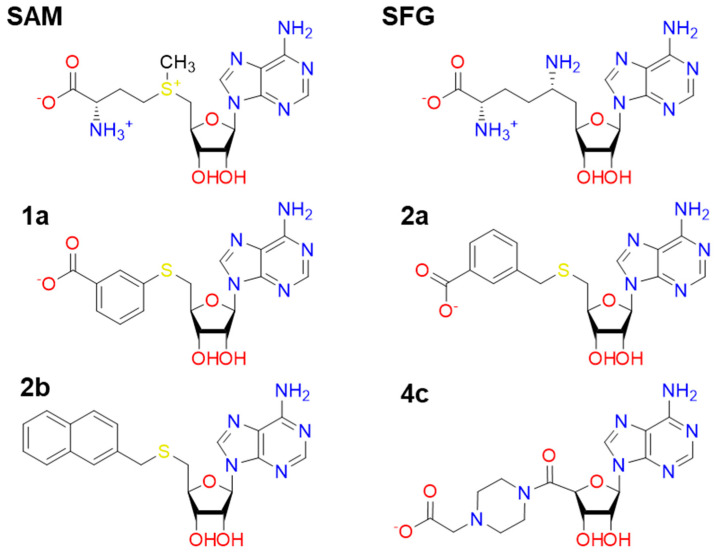
2D structures of SAM, SFG and other analogs (**1a**, **2a**, **2b** and **4c**) synthesized by Bobileva et al. [11].

**Table 1 ijms-23-13972-t001:** Relevant H-bonds between the residues of nsp16 and SAM and its analogs from MD simulations. The labeled atoms are shown in Appendix A.

System	Donor	Acceptor	Average Distance (Å)	Occupancy (%)
SAM	SAM(O3)	Asp99(OD1)	2.68	145.51%
	SAM(N)	Asp130(OD2)	2.82	74.17%
	SAM(N6)	Asp114(OD1)	2.86	55.22%
	SAM(N)	Gly71(O)	2.82	45.40%
	Cys115(N)	SAM(N1)	2.92	15.59%
SFG	SFG(O13)	Asp99(OD1)	2.71	142.09%
	SFG(N22)	Asp114(OD2)	2.87	43.05%
	SFG(N1)	Asp130(OD2)	2.84	25.98%
	Cys115(N)	SFG(N16)	2.91	24.18%
1a	1a(O3)	Asp99(OD1)	2.67	94.61%
	Ser74(OG)	1a(O4)	2.67	70.69%
	1a(N3)	Asp114(OD1)	2.87	65.34%
	Asn43(ND2)	1a(O4)	2.82	21.47%
	Asp75(N)	1a(O5)	2.88	23.41%
	Cys115(N)	1a(N1)	2.92	8.94%
2a	2a(O2)	Asp99(OD2)	2.70	91.07%
	Ser74(OG)	2a(O01)	2.66	65.94%
	2a(N6)	Asp114(OD2)	2.86	43.06%
	Cys115(N)	2a(N1)	2.92	22.34%
	Lys76(NZ)	2a(O03)	2.80	16.14%
	Asn43(ND2)	2a(O01)	2.81	14.43%
2b	2b(O9)	Asp99(OD2)	2.70	82.35%
	2b(N22)	Asp114(OD2)	2.87	38.12%
	Leu100(N)	2b(O9)	2.91	19.86%
	Cys115(N)	2b(N12)	2.92	16.75%
	Tyr132(N)	2b(O5)	2.88	4.13%
4c	4c(O2)	Asp99(OD1)	2.65	161.55%
	4c(N5)	Asp114(OD1)	2.87	41.36%
	Lys76(NZ)	4c(O6)	2.78	12.62%
	Cys115(N)	4c(N2)	2.92	11.74%
	Asp75(N)	4c(O6)	2.90	7.54%
	Tyr132(N)	4c(O5)	2.88	1.67%

**Table 2 ijms-23-13972-t002:** Binding free energy ($\Delta G_{bind}$) values (in Kcal/mol) and their components for the nsp16 systems by MM/GBSA. The experimental free energy ($\Delta G_{exp}$) for SFG and SAM analogs were calculated from IC_50_ values obtained by Bobileva et al. [11].

System	$$\Delta E_{vdW}$$	$$\Delta E_{ele}$$	$$\Delta G_{GB}$$	$$\Delta G_{SA}$$	$$\Delta G_{bind}$$	$$\Delta G_{exp}$$
SAM	−51.2 ± 0.1	−370.4 ± 0.3	358.4 ± 0.2	−6.8 ± 0.1	−70.0 ± 0.1	-
SFG	−45.2 ± 0.1	−83.5 ± 0.3	94.8 ± 0.2	−5.9 ± 0.1	−39.8 ± 0.1	−8.3
1a	−41.8 ± 0.1	72.0 ± 0.7	−71.0 ± 0.5	−6.0 ± 0.1	−46.8 ± 0.2	−10.2
2a	−48.3 ± 0.1	34.8 ± 0.4	−38.8 ± 0.3	−6.4 ± 0.1	−58.7 ± 0.2	−11.5
2b	−51.9 ± 0.1	−54.6 ± 0.3	63.2 ± 0.1	−6.3 ± 0.1	−49.6 ± 0.2	−9.6
4c	−47.8 ± 0.1	98.8 ± 0.5	−74.9 ± 0.3	−6.2 ± 0.1	−30.1 ± 0.2	−5.0

**Table 3 ijms-23-13972-t003:** Residual decomposition analysis of binding free energies for relevant amino acid residues. All values are reported in Kcal/mol. The values for all residues are shown in Appendix A.

AA Residue	SAM	SFG	1a	2a	2b	4c
Asn43	−0.7	−0.3	−3.4	−3.5	−0.9	−1.1
Tyr47	−2.2	0.0	0.0	0.0	−0.2	−0.1
Gly73	−1.8	−1.2	0.0	0.6	0.0	−0.8
Ser74	−1.0	−2.0	−6.2	−4.8	−1.0	−5.1
Lys76	0.1	−0.9	−4.3	−10.4	0.1	−6.2
Asp99	−12.7	−6.1	−5.2	−2.6	−4.0	−8.2
Leu100	−2.6	−2.9	−2.3	−2.3	−2.6	−2.6
Asp114	−1.8	−1.6	−1.3	−1.2	−1.0	−1.0
Cys115	−2.5	−2.9	−2.7	−2.5	−2.4	−1.6
Asp130	−4.4	0.3	2.7	0.7	1.2	2.1
Met131	−2.4	−3.5	−3.0	−2.9	−4.1	−4.7
Tyr132	−2.8	−1.6	−1.3	−2.2	−1.2	−1.7

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
