# Peer review of "Computational Analysis of SAM Analogs as Methyltransferase Inhibitors of nsp16/nsp10 Complex from SARS-CoV-2"

_ijms, 2022, doi:10.3390/ijms232213972_

Round 1

Reviewer 1 Report

Balieiro et al report in silico analysis of binding of analogs of S-adenosylmethionine into SARS-CoV-2 non-structural protein (nsp) 16/10 using MD simulation and explained the basis of their inhibition and poor cell permeability. The work is timely and could be published after addressing the following comments: 

1. Line 65 mentions about “suitable” convergence. I think this and the next sentence should be re-worded to highlight that the system reached an equilibration state. The author already mentions about this later in the same paragraph. 

2. I do not understand what line numbers 87-88 convey. 

3. It would be good to show the ligands in the protein structures in Figure 2.

4. The authors should justify why positive values of electrostatic interactions relate to poor cellular permeability as this was the main basis behind the study. 

5. The manuscript needs a through grammar check before publication. 

Reviewer 2 Report

The authors reported the computational analysis of SAM analogs as methyltransferase 2 inhibitors of nsp16/nsp10 complex from Sars-Cov-2. They studied the structure and the binding energies. They found 2a as the most potent inhibitor with the lowest binding free energy (–58.75 kcal/mol) and more potency. It is interesting, but the following minor revision should be considered. 

1) the change of energies with respect time should give better proof for evaluate if the system is equilibrated.

2) Density functional theory calculations for the binding energies should be more accurate when analysis which one is more potent inhibitor. It will also give more atomic picture of the binding.

3) the text in all the figures are not in consistent format. Some of the text is not clear.
